# Relationship between Bacterial Contribution and Self-Healing Effect of Cement-Based Materials

**DOI:** 10.3390/microorganisms10071399

**Published:** 2022-07-11

**Authors:** Olja Šovljanski, Ana Tomić, Siniša Markov

**Affiliations:** Faculty of Technology, University of Novi Sad, Bulevar cara Lazara 1, 21000 Novi Sad, Serbia; oljasovljanski@uns.ac.rs (O.Š.); sinisam@tf.uns.ac.rs (S.M.)

**Keywords:** microbiologically induced carbonate precipitation, self-healing effect, cement-based materials, concrete innovation, bacterial role

## Abstract

The civil research community has been attracted to self-healing bacterial-based concrete as a potential solution in the economy 4.0 era. This concept provides more sustainable material with a longer lifetime due to the reduction of crack appearance and the need for anthropogenic impact. Regardless of the achievements in this field, the gap in the understanding of the importance of the bacterial role in self-healing concrete remains. Therefore, understanding the bacterial life cycle in the self-healing effect of cement-based materials and selecting the most important relationship between bacterial contribution, self-healing effect, and material characteristics through the process of microbiologically (bacterially) induced carbonate precipitation is just the initial phase for potential applications in real environmental conditions. The concept of this study offers the possibility to recognize the importance of the bacterial life cycle in terms of application in extreme conditions of cement-based materials and maintaining bacterial roles during the self-healing effect.

## 1. Introduction

As a new chapter in human development, enabled by extraordinary technological advances commensurate with those of the previously industrial revolutions, from economy 4.0 has also been expected to bring wide-range achievements in the concrete industry. The concept of “smart concrete” is based on innovations such as translucent, 3D printed, or lightweight concrete, but also self-healing or self-repairing concrete [1]. As the most used material in construction technologies, concrete has annually been produced in an amount of approx. two tons per person on Earth [2]. Due to the high carbon footprint of concrete production (about 8% of the emissions of CO_2_ globally), it is essential to produce more sustainable material [3]. Crack formation is a major problem of concrete matrices, reducing the lifetime of material, as well as increasing repairing costs and anthropogenic influence [4]. Therefore, many investigators and research groups are directed to the creation and application of bacteria-based concrete, with the capacity for self-repairing of cracks on a microscale [5,6,7,8,9]. Although self-healing concrete has been developed for two decades, it is very common that the desired activity of used bacteria does not occur in due time, which represents a problem for further industrialization and commercialization of this type of material [10]. The importance of bacteria contribution during the self-healing (SH) effect has been neglected in many scientific-relevant studies, without explanations of the obtained results in a view of maintaining viability and activity of bacteria in extreme conditions (high alkalinity, limited water availability, minimal nutrient concentration, etc.). Therefore, this review is offering the possibility to understand the importance of the bacterial life cycle in terms of application in extreme conditions of cement-based materials and maintaining bacterial roles during the SH effect. This paper should also serve for monitoring the appropriate characteristics of a bacterium to select bioagents for the SH concept in concrete technology.

The understanding of the relationship between bacterial contribution and the self-healing effect of cement-based materials is presented for the first time from different aspects, including the cruciality of the bacterial activity in the self-healing effect in bacteria-based concrete (Section 2), mandatory requirements for bacteria as bioagents in the concrete technology (Section 3), evaluation of bacterial contribution (Section 4), concrete environment influence (Section 5), and concrete characteristics influence (Section 6) during the self-healing effect.

## 2. Why Is Bacterial Activity Crucial for the Self-Healing Effect of Bacteria-Based Concrete?

The SH effect of bacteria-based concrete can be explained as an effect of autogenously repairing the capacity of material and bio-stimulation of this process by bacterial cells, without any external analysis or anthropogenic intervention [11]. Namely, the bacterial influence during the SH effect is a result of different metabolic activities that lead to carbonate production. The obtained carbonate ions react with available calcium ions from the material, and therefore the formation of CaCO_3_ crystal is inevitable [12]. This process is called microbiologically/bacterially induced carbonate precipitation (MICP/BICP). Theoretically, CaCO_3_ precipitation can occur in environments rich in calcium ions (Ca^2+^), during a metabolic activity that provides carbonate ions (CO_3_^2−^) and enables system supersaturation. Some ubiquitous bacteria with BICP potential have the ability to biocalcify in a wide range of environmental conditions. This phenomenon has been used as a sustainable engineered alternative to conventional techniques in consolidation of soil, sand, stone, and cement, in the improvement of soil characteristics, and in remediation of pollutants in soil, water, and wastewater [10,11,12]. Considering that BICP can lead to effective CaCO_3_ formation, the SH effect of concrete can be accelerated or bio-stimulated using bacteria whose presence and activity may affect crack repairing.

The essential step that enables the SH effect of bacteria-based concrete is the bacterial production of carbonate ions which are transferred from cell to environment. This step also immediately provides an increase in system alkalinity [13]. However, this step is not necessarily the most important role that bacteria can play in CaCO_3_ formation. BICP is consequently a result of metabolic activity, CaCO_3_ nucleation, and crystal formation around individual bacterial cells. Besides bacterial contribution to changes in ion concentration and pH value, active bacterial cells have the potential to be efficient nucleation centres during CaCO_3_ formation and repairing cracks. A high concentration of bacterial cells in cracks will also contribute to a higher degree of nucleation and rate of repairing [14]. Theoretically, a number of viable and metabolic active cells in a concrete matrix represents a number of nucleation centres [15]. As shown in Figure 1, BICP can also be viewed as a reorganization of ions present in the cell microenvironment and extracellularly produced ions. It should not be forgotten that crystal morphology will continue to be conditioned by the concrete environment in which precipitation initiates [12]. In their latest research about the mechanistic concept of BICP, Hoffmann et al. [16] reported that the large cell surface area to volume ratio can be a positive moment for the SH effect, due to the fact that the cell wall is covered by negatively charged area; however, some essential differences between Gram-positive and Gram-negative bacteria need to be pointed out. Briefly, the negativity of Gram-positive cells is primarily imparted by carboxyl and phosphate groups of teichoic acids, while the same role has phospholipids and lipopolysaccharides in the cell wall of Gram-negative cells. The presence or absence of S-layer glycol proteins with glycosylated long carbohydrate chains can also have further influence on the negativity of the cell surface area, with higher dependence on prominent structural groups. Although all bacteria can have BICP potential, Gram-positive bacteria are more often the first choice for engineering applications.

The result of the SH effect in bacteria-based concrete is a chemical reaction between Ca^2+^ and CO_3_^2+^ ions in the bacteria microenvironment, with the formation and maturation of CaCO_3_ crystalline on an electronegative cell surface. This would confirm the previously stated fact that bacterial cells are attributed to the BICP effect as nucleation centres. In summary, bacteria refer to inevitable changes in the system, increasing the reactivity of the resulting metabolite to Ca^2+^ ions, as well as accumulation of all other ions on a cell surface [17].

## 3. Mandatory Requirements for Bacteria as Bioagents in Bacterial Concrete Technology

Due to the fact that BICP is widely used in various engineering techniques, the selected bacterial culture, in addition to meeting preconditions of an efficient BICP process, must be safe for use in all environments. Vidaković et al. [18] emphasized that the selected bacterium must not pathogenic before, during, or after the bioprocess. Briefly, the nature of the viable agent bacterium has a great impact on the engineering needs. From a practical perspective, microbial pathogenicity is the capacity to cause disease during manipulation and application with bioagents. The great majority of microorganisms used in bioprocessing are harmless and many are beneficial. Whenever possible, “generally recognized as safe” or GRAS species should be used in commercial processes. The use of new species or those with unknown risks must be preceded by an assessment of risk, including pathogenicity testing. Understanding how viable cells spread and use substances in their surrounding are essential to assessing the risk of further processing. Considering a request for suitability for use in any engineering approach, risk assessment of using bacteria in concrete has to involve:Complete characterization (taxonomic identification, history of bacterial species/strain in terms of recognized pathogenicity, genetic changes, ecological properties, etc.);Methods for reproduction and application of the bacterium, which are in accordance with biosafety laws (e.g., antibiotic resistance, genetic mutations, etc.);Defining stability, activity, packaging, and storage of a biological agent in compliance with legal regulations;Comprehensive assessment of unpredictable environmental (and health) risks for release of (genetically modified) bacterium into the environment [19].

## 4. Evaluation of the Bacterial Contribution to the Self-Healing Effect

### 4.1. Metabolic Activities That Lead to Carbonate Ions Production

CaCO_3_ precipitation is a lateral effect of metabolic activities such as ureolysis, denitrification, ammonification, sulphate reduction, methane oxidation, i.e., in all effective metabolic reactions in which extracellular release of carbonate ions occurs and causes pH changes [12]. Figure 2 shows metabolic pathways which have an engineering potential towards efficient carbonate precipitation in the concrete matrix. This occurrence can take place under anaerobic conditions when it is most commonly attributed to methane oxidation, sulphate reduction, and denitrification. Anaerobic oxidation of methane favours carbonate precipitation because much of the methane is converted to CO_2_, especially in marine sediments [12].

Furthermore, abiotic dissolution of gypsum will provide an environment rich in sulphates and calcium ions, with sulphate-reducing anaerobic bacteria efficiently generating H_2_S and releasing bicarbonate and carbonate ions. Among anaerobic processes, denitrification, which involves nitrate reduction to nitrogen, is the most efficient metabolic activity for BICP. Adopting nitrates, denitrifying bacteria increase the pH value due to the utilization of hydrogen ions and creation of carbonates, which results in supersaturation [20]. The by-product of complete denitrification is molecular nitrogen, which is not harmful and does not affect the further growth, and precipitation of the formed crystalline forms. However, if the denitrification process is not complete, the accumulation of toxic nitrites and other nitrogen oxides which cause the greenhouse effect and ammonia is possible [18]. Additionally, examining bacterial connection to oxygen would provide an answer to the question of whether the bacterium would have the same SH effect on the surface, sub-surface, and the depth of the concrete matrix being treated.

In view of simpler application in engineering processes, aerobic metabolic processes, such as amino acid ammonification and ureolysis, are far more desirable. They not only represent a simpler process, but also give a far greater chance to study. Ammonification involves amino acids decomposition into carbonate and bicarbonate ions. Subsequent ammonia hydrolysis accumulates hydroxyl ions which cause system supersaturation [12]. Ureolysis is one of the most efficient aerobic reactions mediated by BICP [12,21,22]. In presence of urea and calcium, ureolytic bacteria are highly efficient inducers of CaCO_3_ precipitation since they produce ammonium and carbonate ions quickly, alkalizing a medium and enabling rapid precipitation in concrete [23,24]. When talking about bacteria with BICP potential, their biodiversity in different ecosystems and sources should also be mentioned. This microbial community successfully modifies the geochemical conditions of the environment and strongly affects matrix permeability and alkalinity. Differences in CaCO_3_ precipitation effectiveness, and morphology and phases of crystals, can be a basis for selection of bacterium as an effective agent in BICP [25].

#### How to Find an Appropriate Bacterium for Self-Healing Concrete?

A review of the scientific literature shows that interest in isolating natural strains of bacteria with BICP potential in the SH effect of cement-based materials has expanded in the last ten years. On the other hand, an identical scheme of isolation and selection has rarely been followed. Random isolation schemes vary from the ultimate goal of selection, and differ both in the initial steps of sample processing, and further manipulation and application. Isolation sites for bacteria with BICP potential are closely related to ecosystems in which a special contribution of these bacteria is expected. Namely, the largest number of bacterial strains with BICP potential was isolated from alkaline and calcifying soils. It is noticeable that isolates belong to the genera *Bacillus* [26,27,28] and *Sporosarcina* [26,27,29]. As representatives of Gram-positive bacteria, species *B. lentus*, *B. subtilis*, *B. megaterium*, and *B. thuringiensis* stand out, as well as *S. pasteurii*, *S. globispora*, *S. ginsengisoli*, and *S. koreensis*. In addition, strains of *Lysinibacillus* genus have often been isolated [29,30,31,32]. Among bacterial isolates from aquatic systems and calcite sludges, *Bacillus* species stand out, but *Pseudomonas*, *Salinivibrio*, *Halomonas* [33], *Paenibacillus* [34], *Acinetobacter*, and *Pseudomonas* isolates [35] have also been reported. Al-Thawadi and Cord-Ruwisch [36] isolated only *Bacillus* strains from sedimentary sludge. closely related to aquatic systems is isolation from sand, where the presence of representatives of *Bacillus*, *Paenibacillus*, and *Sporosarcina* have been confirmed [37,38].

A significant number of bacteria with BICP potential from sediments, as well as different natural and cement-based materials, have been identified (Table 1). The bacteria were isolated from caves, calcite quarries, mines, and quarries, but also from the seabed and the surface of stones present in calcite soil. Furthermore, *Bacillus* species are predominantly present in samples of cement, mortar, limestone, and concrete, but are also isolates from surfaces of historical monuments, historical limestone, cement substrate, and painted mural layers. For application in cement-based materials, reference strains are often found, which is either a working microorganism or a relevant strain for comparing the obtained results (Table 1). Isolation of natural strains of bacteria opens possibilities of genetic modification to increase carbonate production through a higher rate of metabolic activity. This is a way to achieve fast and efficient CaCO_3_ precipitation, so Begdale et al. [39] used ureolytic genes of *S. pasteurii* to create genetically modified bacteria *P. aeruginosa* MKJ1 and *Escherichia coli* MKJ2. On the other hand, Achal et al. [27] treated soil isolate *S. pasteurii* with UV light to be more resistant to higher pH values and significantly increase metabolic activity during BICP.

### 4.2. Keeping Bacterial Viability—The Key to Effective Self-Healing Effect

All mentioned bacteria are considered to have a special place in nature since they have the ability to generate an alkaline environment during carbonate production [25], and consequently during the SH effect of cement-based materials. Moreover, their resilience is reflected in the possibility to survive in stressful environments such as cement-based materials [64]. Precisely, when exposed to drastic changes in the environment (in alkalinity, water availability, external stress, or/and to unfavourable living conditions), some bacteria can increase their survival rate by forming a protective, resistant form of life called spores [65]. Various environmental factors in concrete can affect bacterial activity by reducing proliferation (growth, development, and replication) and inducing the process of spore formation—sporulation (Figure 3). Changes in temperature, pH, oxygen presence, and mineral concentration are known to affect sporulation, but the most important inducers of this process are nutrient deficiency and microbial population density. By transitioning from a proliferative way of life to the latent phase (Figure 3), bacteria limit metabolic activities and store genetic material until favorable conditions are achieved again [66].

Spores are metabolically extremely dormant, and this dormancy represents excellent resistance to many factors related to cement-based materials, such as heat/cool cycles, dry/wet cycles, radiation, and chemicals, and therefore their survival over a longer period is expected [66]. It is important to emphasize that spores cannot directly participate in the SH effect of bacteria-based concrete. Although spores are latent, they are still interacting with the external environment, which allows them to pass into vegetative cells after only a few seconds of favourable living conditions and induce BICP after a certain time in a concrete matrix [67].

As well as adaptation to high pH values, the possibility of the formation of spores indicates high resistance, which is preferable in cement-based materials [68]. Consequently, sporogenic bacteria are always a better choice for the SH effect of cement-based materials compared to asporogenic bacteria. Therefore, sporogenic bacteria isolated from extreme environments are an exceptional biotechnological “tool” for advancing engineering approaches to SH of cement-based materials. Through increased tolerance and survival in variable living conditions, high metabolic activity with minimal nutrient content, and the possibility of subsequent cell activation, the use of sporogenic bacteria leads to the prevention of economic losses in long-term technological processes [69].

To keep bacterial viability during unfavorable conditions in the concrete matrix, protection of bacterial cells is necessary. At first, some researchers practiced the addition of sporogenic bacteria directly into cement paste, but they quickly realized that the number of active cells was reduced to a minimum due to the influence of unfavourable environmental conditions [7,56]. Briefly, adding bacteria and nutrients directly to the concrete matrix can improve some mechanical properties, but bacterial survival was between 1.9 and 7% after 10 days of curing [70]. This could have occurred due to different environmental impacts, such as long-term stress on cells which started immediately after addition to concrete, but also might be a consequence of the unpredictable and highly variable hydration of cement. The water unavailability and the reduction of free water in a material system due to cement hydration changes may be the most significant influence on the minimal survival rate of bacterial cells that are in direct contact with the concrete. Nevertheless, in the past decade, many self-healing concretes have been developed with encapsulated bacterial cells, which represented the best opportunity for bacteria to remain viable, active, and protected for a longer period in a concrete matrix. The encapsulation of bacterial cells and addition of prepared capsulates into concrete have been investigated by various methods, including different factors such as characteristics of cement-based materials, type and concentration of bacteria, material, size, distribution and quantity of capsules, etc.

The formed capsulation system needs to be inert in contact with bacterial cells, strong enough to resist the concrete mixing process, must not negatively influence the mechanical properties of concrete, and be fragile enough to open whenever cracks appear [71]. The encapsulation process has included various materials for capsules, such as diatomaceous earth [72], lightweight aggregates [73,74,75,76], ceramsite [77], silica gel [78], hydrogel [5], graphite nanoplatelets [74], melamine [79], etc. In the mentioned research, different bacteria were used: *Bacillus sphaericus*, *B. alkalinitrilicus*, *B. cohnii*, *B. subtilis*, *S. pasteurii*, *B. mucilaginous*, etc. The best results of adding encapsulated bacterial cells were achieved using expanded clay and melamine for encapsulation of *B. cohnii* and *B. sphaericus*, respectively, which resulted in the healing of cracks of 0.79 and 0.97 mm [61,76]. Additionally, lightweight aggregates had the best ratio between the positive influence of compressive strength of concrete and maximum crack healing rate, and represented the most promising technique for the encapsulation system of bacterial self-healing concrete. Regarding an encapsulation system for this application, it is very important to assess the different aspects of using capsules in concrete in view of bacterial activity, concrete characteristics, economic and ecological cost-effectiveness, etc. The encapsulation system in bacterial self-healing technology provides the required activity of bacterial cells during crack appearance and potential effectiveness under many damage measures, but can be complex in regard to casing and difficulties in releasing a healing agent can appear [80].

Further, it is noticeable that scientific groups related to the creation of bacteria-based concrete have used different bacterium, but also different initial concentrations of cells, often without optimization of this parameter during primary laboratory testing. Table 2 shows different levels of commitment to the used bacterium, as well as a wide range of different methodologies for monitoring three essential parameters of SH concrete:Rate of bacterial activity (almost no one pays attention);Level of SH effect (monitoring with standardized and/or non-standardized, but well-known methodologies in concrete characterization protocols);Characteristics of concrete (monitoring with standardized and/or non-standardized, but well-known methodologies in concrete characterization protocols).

Although the crack-repairing effect has been established in all selected research articles, it is difficult to compare results among authors due to different experimental setups and used methodologies. Different experimental setups are a consequence of the absence of universal standard methods with good practical settings for the evaluation of SH bacteria-based concrete performances. Bacterial contribution to the SH effect of a concrete matrix is based only on a comparison of differences between control and bacteria-treated samples, while the monitoring of bacteria role, activity, or viability was not defined in almost all mentioned papers. It can be summarized that many investigations related to bacteria-based concrete lack microbiological and biotechnological points of view. As a multidisciplinary approach in the field of a new generation of concrete is required, this can be an appropriate critical point for further optimization of bacteria-based SH concrete. Consequently, reducing the large costs of SH concrete implementation can be strongly associated with optimization of the number of bacterial cells, the amount of essential nutrients, determination of bacterial behavior and contribution to the SH effect, etc.

### 4.3. Physical Contribution of Individual Bacterial Cells to Self-Healing Effect

Each cell can additionally contribute to the SH effect through physical predisposition. In particular, this role of bacterial cells in the SH effect is strongly associated with the potential of active cells to be CaCO_3_ nucleation centres. The connection between the physical contribution of cells and the SH effect can be based on:Cell geometric compatibility (cell size, specific surface area, and volume) and motility;Cell−surface electronegativity and hydrophobicity (affinity, types, and amount of chemical bonds on the cell surface);Cell membrane permeability (in a function of extracellular carbonate production rate);Biofilm production (possibility and rate of adhesion on inorganic surfaces) [88];The cell size can be a crucial parameter for bacterial activity in view of variable concrete porosity during dry/wet cycles in material [7]. Figure 3 also showed a correlation between average cells and pore sizes during hydration and volumetric changes in concrete. Besides water availability, pore size changes can also be a limiting parameter for spore activation and cell proliferation. Based on cell size, cell−specific area and volume of active bacteria that are involved in BICP are between 2.6 and 8.55 µm^2^ and between 0.3 and 1.64 µm^3^, respectively [88]. Considering that bacteria require availability in space, pore volume can influence bacterial activity, and as a result, CaCO_3_ precipitation.

Except for bacterial size, effective calcium accumulation is going to depend on the cell surface characteristics. The cell surface is a place for CaCO_3_ nucleation because of its cell electronegativity and hydrophobicity. These characteristics of the cell surface originate from the considerable number of chemical bonds present on cell walls [89]. Strong electrostatic affinity for cation accumulation in a certain place enables the precipitation of CaCO_3_ crystals [90]. It is important to note that the binding of cations to cell wall structures begins in a slightly acidic environment (Figure 4). However, functional groups of cell membranes, such as carboxyl, hydroxyl, and phosphate, become very reactive under highly alkaline conditions, creating a strong electrostatic affinity for calcium attraction and accumulation on cells [17]. During CaCO_3_ crystal accumulation around electronegative cells (Figure 4), reduction of electronegativity and hydrophobicity is to be expected. Several research groups have measured these parameters during BICP, and reported that electronegativity can be neutralized with completely CaCO_3_ crystal-surrounded cells [24,88]. When CaCO_3_ deposits are formed around a whole specific cell area, it is a significant question whether that cell is a further active participant in the SH effect.

There are two assumptions about what happens in this situation: spore formation (unfavorable conditions are achieved) or cell death (complete stopping of viability). Neither of these two has been proven, but both can be a reason for the absence of a long-term SH effect when the initial number of cells in concrete is low. Therefore, it is important to initially provide a high number of active cells in a concrete matrix [11]. For example, Zhang et al. [86] defined that the optimal concentration is not less than 10^8^ CFU, but the SH effect will be initiated with 10^5^ CFU. Any of the research groups mentioned in Table 2 did not define the optimal number of bacteria in self-healing concrete. The number of active cells is a limiting factor, given that, due to unfavorable environmental conditions, sporulation occurs and the level of system reactivity decreases. In addition, many cells remain trapped in pre-formed crystals, and the method of application and monitoring of bacterial activity during the BICP process is very difficult [91]. Additionally, cell membrane permeability can direct the SH effect in the same way, because CaCO_3_ accumulation can affect ion transport mechanism functions, and consequently, carbonate productivity, which is in direct function of membrane permeability. The obtained crystals around cells can cause partial or total disruption of extracellular carbonate production. The SH effect in a concrete matrix can also be affected by extracellular polymeric substance (EPS), resulting in cell adhesion and biofilm formation during bacterial activity. Hoffman et al. [16] have suggested that the answer to the question of whether the precipitating bacterial cell is alive or dead after the SH effect differs at a bacterial species level, but also did not exclude the possible differences due to environmental conditions and the methodology of applying the cells in the system.

Figure 5 explains bacterial behavior with or without biofilm formation on the concrete surface during the SH effect. The biogenic polymer matrix allows a large number of bacterial cells to attach to the substrate and/or surface of artificial materials [12]. Moreover, EPS consists of charged and neutral polysaccharide groups that not only bind to surfaces, but also serve as an ion exchange system for “entrapping” and accumulating ions from the environment [16]. This is correlated with the negatively charged cell wall (explained in Section 2), and EPS surfaces tend to be negatively charged and have a high affinity for cationic species such as calcium ions. Carboxyl groups in particular have been found to contribute strongly to the metal-binding capability [16]. Consequently, the EPS matrix has a high total electronegative charge, and calcium and nutrient accumulation will be simplified and localized near biofilm cells. However, this structure may also inhibit the formation of CaCO_3_ deposits, and it is necessary to degrade EPS to begin the SH effect [50].

## 5. Influence of Concrete Environment on Bacteria-Based Self-Healing Effect

The bacteria-based SH effect is often influenced by variable environmental conditions in cement-based materials. After choosing a metabolic pathway that will mediate the process in terms of efficient extracellular production of carbonate ions, as well as investigation of all potential impacts of bacteria during the SH effect, the potential for bacterial survival in adverse environmental conditions is also very important [90].

### 5.1. pH Value

Concrete represents an alkaline environment, but is very changeable during its lifetime. Namely, the pH value of ordinary Portland cement concrete varies between 12.5 and 13. Such a high value can be decreased over time by different deterioration mechanisms such as chloride ingress, carbonation, acid attack, or the carbonation process. These formations can reduce the pH value of concrete to values less than 9 [92]. The metabolic activity, especially that which is enzymatic, will depend on pH value and its changes. The high pH value favors the generation of carbonate and bicarbonate ions, so the BICP process is often induced in alkaline environments. The alkaline by-products are obtained during carbonate saturation so that the optimal pH value for metabolic activity and SH effect is often the same [90]. Achieving a pH value above 9.5 provides optimal conditions for the successful maturation of stable crystalline forms such as calcite and aragonite [17], but a consequence of CaCO_3_ precipitation can be a pH decrease [93]. Gat et al. [94] proved that bacteria can change pH value by almost 2 pH units during CaCO_3_ precipitation, with rapid and abrupt changes towards a highly alkaline matrix at the very beginning of the SH effect. This would mean that alkalotolerant or alkalophilic bacteria are more preferred in the SH effect. Alkalophilic bacteria have optimum growth at pH values around 10, showing satisfactory BICP potential in the SH effect [88,95].

### 5.2. Presence, Concentration, and Availability of Calcium Ions

Concrete represents a matrix rich in calcium, but not all calcium is available for the SH effect in materials. Namely, a very low concentration of calcium ions is present in extracted pore solution compared with calcium ions in the alkali−silica reaction in concrete. One of the inevitable phenomena in cement-based materials is calcium leaching, which represents a degradation mechanism through a progressive dissolution of hydrates and migration of calcium sources, especially to pores [96]. Calcium and other ions are transported through pores, cracks, and all other physical damages due to water diffusion. Many of them are highly reactive and strongly bound to solid particles in a concrete matrix [97]. Calcium leaching and cracks in concrete often occur at the same time. Therefore, in concrete pores, a very low concentration of free calcium ions is expected, which might be suitable for bacterial activity. In view of the SH effect in cracks, calcium availability in (old) cement-based constructions enables local repair, without the initial addition of calcium as a necessary substance for the BICP. However, high concentrations of calcium can limit bacterial activity by the substrate inhibition mechanism, so values up to 0.25 M of free calcium are considered the best choice for an effective SH effect [48]. Furthermore, sporulation in a medium rich in calcium was not observed in the case of *Bacillus* strains [98]. By fixing potentially harmful calcium ions, further bacterial activity and maturation of crystalline forms formed around cells are enabled at the same time [41]. Although calcium ions can be available in concrete, almost all research groups practice adding calcium sources as part of the essential nutrients for the targeted metabolic activity of bacteria. For example, calcium chloride is the primary choice for ureolytic activity, which is far more intense in the presence of this salt compared with other inorganic calcium-involved salts. One of the reasons for calcium chloride efficiency is its high solubility. The result of this, however, is that abundant calcium hydroxide forms, and it seems that it is this calcium hydroxide that is the source of calcium used by the bacteria [99]. However, recent research has suggested that this can carbonate and be unavailable for future reactions, and delay or stop the SH effect [100]. In the case of the process of oxidation of amino acids, the most commonly used salts are in the form of calcium lactate, calcium nitrate, calcium formate, and calcium acetate [80]. However, the choice of the calcium source will depend on bacteria type, conditions of use, the material matrix and its influence on final mechanical characteristics, application and use conditions, etc. Some authors, such as Tziviloglou et al. [101], suggested the encapsulation of all nutrients including calcium ions (i.e., calcium lactate and yeast extract) in lightweight aggregates, but this can strongly influence the availability of calcium ions after crack appearance, and also impact the economic parameter of the healing agent. On the other hand, for the commercialization of healing agents for the SH effect in cement-based materials, it is necessary to form a stable product, which could include the encapsulated calcium and essential nutrients, which in that case would be added as a dry additive to concrete.

### 5.3. Presence of Other Elements, Cofactors, and Inhibitors of Bacterial Activity in a Concrete Matrix

Although 80% of the composition of concrete is normally aggregate and has a large mineral composition that is unreactive, some of the components in (non)hydrated concrete can have a stimulative or inhibitive effect on bacterial activity. The influence of different ions in concrete during the SH effect can be two-fold. Namely, the chemical composition of concrete is variable and depends on the raw materials, but usually consists of calcium silicates, aluminates, and ferrites, in combination with calcium, silicon, aluminium, magnesium, manganese, titanium, sodium, potassium, sulphur, and iron, in forms which react with water [102]. From a bacterial activity point of view, the presence of Na^+^, Mg^2+^, K^+^, Cl^−^, Br^−^ does not affect bacterial activity, whereas the presence of Mn^2+^ may cause a decrease in bacterial activity and increase the sporulation effect. Bacterial activity can be improved, but is not completely dependent on the presence of Li^+^, Zn^+^, and Ba^+^ ions [103]. In addition, TiO_2_ has a well-known killing effect on bacteria cells, and its presence in a material can have a biocide effect on numerous bacteria [104]. On the other hand, the presence of nickel ions can positively influence the functionality and structural integrity of the ureolytic enzyme [105]. In summary, an examination of the relation of bacteria and chemical structure elements, especially potential inhibitors of bacterial viability, need to be established to assess bacterial behaviour during the SH effect in concrete. Considering that cement-based materials present a very poor medium in the view of nutrients for bacteria, the addition of essential nutrients is required, regardless of the primary components of concrete.

### 5.4. Presence and Availability of Essential Nutrients for Metabolic Activity

Concrete represents an extremely harsh environment for bacteria due to its low nutrient availability [106]. Therefore, providing the main substrates that trigger targeted metabolic reactions and bacterial activity is crucial for an efficient self-healing effect in a concrete matrix. Optimizing the concentration of essential nutrients, which are a source of carbon, energy, and/or nitrogen, will be a key factor in achieving maximum CaCO_3_ precipitation and the self-healing effect [101]. For example, DeJong et al. [107] proved that the optimized urea content, as an essential substrate during ureolytic activity, leads to uniform CaCO_3_ precipitation in a short period. If the main nutrient accumulates in the system of nutrients are not available for bacteria, the metabolic activity can be inhibited, but also if there is a deficiency of calcium ions, supersaturation will be absent [108]. Most of the literature related to self-healing concrete has used the encapsulation of nutrients in a similar way to that of bacterial cells, and is often practiced during the formation of self-healing concrete. However, the optimization of nutrient content in concrete, the method of its application in a matrix (nutrients are soluble in water, need to be available at the moment of crack appear, have to be uniform in the whole matrix, do not induce bacterial activity at the initial curing phase, etc.), the impact of concrete characteristics (can increase/decrease mechanical properties), and the possibility of used secondary sources (pure chemicals are economically and environmentally unprofitable) need to be addressed before application and commercialization of bacterial self-healing concrete. It is estimated that bacterial cells and nutrients required for the self-healing concrete preparation are up to 80% of the total operating costs [109], and thus, the advances in an alternative inexpensive nutrient source are essential to support the widespread use of the bacterial-based concrete. Replacement of urea, calcium salts, yeast extracts, and similar nutrients has been done with some industrial by-products, including lactose mother corn steep liquors [110], perilla meal, rice bran, sesame meal, soybean meal, soybean pulp, wheat bran [111], and waste-derived polyhydroxyalkanoate [112], etc.

### 5.5. Other Influence

Some additional influences of the cement-based material environment on bacterial activity and the SH effect can be summarized through the presence of chlorides and sulfates, but also the carbonation process in the concrete matrix. Cracks in marine environments appear in concrete due to penetration of chloride and sulfate ions. The chloride concentration in the cracks is similar to that on the outer concrete surface exposed to the chloride environment. The repercussion of this process can be the promoted corrosion of rebars and the destructive process of the infrastructure unit [113]. Therefore, a special approach needs to be adopted for application of the SH concrete in a chloride-rich environment, and selection of the appropriate bacterium is required. In view of bacterial life in these conditions, it is necessary to ensure the use of a bacterial strain which is tolerant of a high amount of salt during the SH effect. Some bacteria with a strong precipitating effect are reported as highly tolerant strains to the chloride present, and belong to *Bacillus* genera [114,115]. The situation is similar regarding sulphate attack on the concrete matrix. Using the appropriate bacterial culture, sulphate resistance of bacterial-based SH concrete can be improved for 120 days [100], but the bacterial concentration and other requests for improving sulphate resistance have not been sufficiently investigated. In real-time environments, a concrete matrix is subjected to dry/wet cycles, and carbonation can occur before cracks appear. After carbonation, calcium ions are in a less soluble form, and bacteria cannot utilize targeted ions [116]. Some evidence, especially for historical samples where the carbonation rate is high [117], indicates that the bacterial SH effect may not occur if optimal amounts of nutrients (with calcium ions) are not directly added to the system or on the surface of the cracks.

## 6. Influence of Concrete Characteristics on Bacteria-Based Self-Healing Effect

Concrete is a specific, multicomponent and polydisperse matrix that is obtained by homogenizing a mixture of aggregates, cement, water, and additives. The characteristics of concrete are generally a function of an extremely large number of different influencing factors, such as characteristics of the applied components, qualitative ratios of the components in the mass of concrete, many technological factors, etc. [118]. However, despite a large number of mechanical loads and environmental actions that affect concrete characteristics, four are especially important for bacterial activity and the SH effect:Permeability (due to the water availability);Porosity (due to space limitation);Crack size (due to contact with the environment and collection of nutrients, water, and oxygen);Aging rate.

These characteristics can be explained through the concept of limiting factors for bacterial activity in some media. Namely, when a bacterium is under stable conditions, the essential factor existing in quantities closest to a critical minimum needed for that bacterium will tend to be a limiting factor for growth and activity [119]. This concept is well-known to microbiologists and ecological engineers, but can also be base for a better understanding of the bacterial life cycle during the SH effect of concrete.

### 6.1. Porosity, Permeability, and Water Availability

Concrete contains numerous pores of broadly varying sizes, from several nanometres to several millimetres. Variations in space are a consequence of the intrinsic heterogeneity of pore sizes in concrete [120]. As a measure of void volume in concrete, porosity is a special factor that can limit bacterial activity. In Section 4.3, the importance of average pore size in concrete, related to bacterial size and the sporulation process during BICP, is explained. The pores in concrete represent a void of a certain size where moisture can transfer, and almost all pores are interconnected and enable the diffusion of all soluble components, as well as the distribution of bacterial cells [121]. If it is necessary to connect porosity and permeability, it can be said that more porous concrete can have a high rate of permeability, but that does not have to be the rule. Small or unconnected pores reduce permeability, but also a chance for bacterial activity. Furthermore, the permeability of mature and well-cured concrete matrix can be extremely low, while porosity can be high (a large share of unconnected pores). The relationship between micropores and ions is not a simple process and cannot be explained with diffusion or advection, because phenomena differ in the rate of pore saturation with available water [122]. Briefly, diffusion may not occur if water fills concrete pores, due to the complex interconnection between pores and physical phenomena which follow this situation. The presence of water in pores, irrespective of their sizes, is very important for the MICP process to happen [123], but is not necessarily a source of free water for bacterial activity.

Table 3 summarizes the effects of bacterial activity in self-healing concrete on permeability, porosity, as well as crack size, which was healed in the established conditions during experiments. The permeability, as the rate of flow of moisture through concrete under a pressure gradient, always decreased, regardless of bacteria type or number of bacterial cells in concrete. The decreased rate varied from 12% to 94%, which suggests that after the initial SH effect, permeability and porosity are on a lower level, and further bacterial activity is minimized. This could be a reason why the initial phase of the self-healing system is more effective than the later phases in many research articles [8,124,125]. The direct connection between permeability and water availability strongly influences bacteria, since all bacteria require high humidity levels for growth and activity. Decreasing water availability in materials will decrease the possibility of the activation of cells and the SH effect.

### 6.2. Crack Appearance and Size

Cracking in concrete is very usual and affects the lifetime of a structure, but also enables a higher influence of environmental actions. Cracks provide an additional source of water and waterborne ions, which can have an essential role in the SH effect through the activation of bacterial cells in cracks. Besides water, cracks can be filled with different fluid phases, for example, air, salt, acid solution, etc. [132]. Crack size can limit the efficiency of the SH effect, which is represented in Figure 6. If the crack width is very big, the SH effect cannot be deployed through the whole crack, because bacterial cells do not have enough capacity to induce the required CaCO_3_ amount. The same number of bacteria in a narrower crack could be sufficient for the full SH effect. This is the critical point where the optimization of the number of bacterial cells and CaCO_3_ productivity are necessary. As shown in Table 3, the SH effect is achieved in wider cracks if a higher number of bacterial cells are present. With a complete filling of cracks, the bacterial life cycle cannot be monitored, not even indirectly, until a new crack opens and the SH effect is reinduced. It is essential to emphasize that all the studies measured a healed crack in terms of width, while the real crack geometry is very complex and includes tortuosity, roughness, and depth, which also may have affected the SH efficiency. On the other hand, the decrease in pores and crack size also made transportation of calcium ions more difficult [8], which also influenced the SH effect. Van Tittelboom et al. [126] emphasized that the average rate of crack filling in bacteria-based concrete is 15 μm/day, but it can be added that this SH rate will depend on many factors which cannot be controlled at the same level during crack healing.

Figure 6 also emphasizes one more crucial difference between theoretically and experimentally obtained results in the SH concrete field. Briefly, in many cases, the SH effect ended with the sealing of the crack making some kind of bridge between crack edges. In this situation, the SH effect can be stopped since the formed sealing deposit prevents the ingress of oxygen and leads to an oxygen deficit deeper in the crack, and bacterial activity stops [133]. Tan et al. [134] suggested that the healing products were mostly generated on the surface of the cracks, with the partial formation of dense healing products (loose particle-like crystals) in pores and voids. The hypothesis that SH can be triggered in a deeper area of cracks after the sealing effect is also stated in other scientific investigations. The potential solution of this challenge can be found in the reduction of initial efficiency of the SH system, or microbial healing coupled with an inorganic method to obtain healing at a deeper depth of the crack [113,135].

### 6.3. Age Rate

The aging of concrete is an inevitable process and usually begins to appear in individual elements of the structures, leading to nonuniform behavior and cracks appearing [136]. In view of bacterial activity, some investigators consider that the bacteria-based SH effect is convenient for cracking at an early age of concrete. Consequently, a hypothesis is that the SH effect decreases with the aging of concrete structures [124]. According to one group of scientific studies, the obtained results of crack monitoring indicated that the SH effect is better in the first 28 days, and crack filling slows down after that [137,138,139]; however, this phenomenon can be related to the autogenous capacity of cement-based materials. In this situation, a question arises as to whether the material lost the autogenous capacity, or whether bacterial cells were not active for further bacteria-based SH effect. Indeed, some investigators believe that the concrete older than 60 days lost its SH effect based on bacterial activity because of the initial activation of bacterial cells and their loss over time [8]. On the other hand, during a recent investigation of the capability of cement-based materials to re-heal previously healed cracks, Justo-Reinoso et al. [140] proved the repeatability of the bacteria-based SH effect after 22 months in the same concrete samples, but not at the same place where the previously cracks appeared. This is proof that the active bacterial cells were not available within the primary crack area due to a reduction in number after the initial SH effect, but the concrete matrix contained viable encapsulated bacteria at the different positions and the new cracks were healed with the average healing ratio of 93.3%

## 7. Concluding Remarks

The use of self-healing bacteria-based concrete certainly represents an innovative and cutting-edge issue for the sustainable cement-based materials industry. Considering that many research designs of this concrete have disregarded the microbiological view of the SH effect of bacteria-based concrete, a major recommendation is that one of the laboratory steps include an assessment of the microbiological aspect. The first step is essential and has to include a selection of best-performing bacteria for the SH of cement-based materials. The selection of bacteria will depend on the available source isolates from nature, purchases from a national collection of microorganisms, genetic modification, etc. An ideal bacterium for the SH effect of cement-based materials has an efficient metabolic activity that leads to the production of carbonate, tolerance to the alkalinity of concrete matrix, high survival, and rapid activation of cells in concrete, but also has a physical predisposition for the effective SH effect. Bacterial contribution in the crack filling, decreasing the possibility for the formation of wider cracks, and reducing the permeability and porosity, leads to an increase in stability and a longer lifetime of the concrete. Therefore, the optimization of using bacteria in concrete is necessary for reducing additional costs and creating more economic materials.

Perhaps the biggest potential problem during the creation of SH concrete is obtaining homogeneous treatment and monitoring processes in real conditions. Successful implementation requires upgrading the system established at the laboratory level. Unlike laboratory conditions where most parameters can be controlled, it is necessary to monitor multiple variables in in situ treatment, both during the application of SH concrete and after/during the cracking process. These variables have to include the monitoring of microbiological activity. The possibility of the complete stop of the self-healing effect over time can be expected due to the loss of a bacterial role in crack repairing. The reason for this is the dependence of bacterial activity on a large number of environmental factors (pH, temperature, water availability, the porosity of materials, nutrients, metabolite diffusion rate, etc.). A design in which it is possible to monitor and influence microbiological activity in a large system is a difficult task, not only because of geological accessibility, but also because of the heterogeneous distribution within the system, as well as the lack of standardized monitoring methods.

## Figures and Tables

**Figure 1 microorganisms-10-01399-f001:**
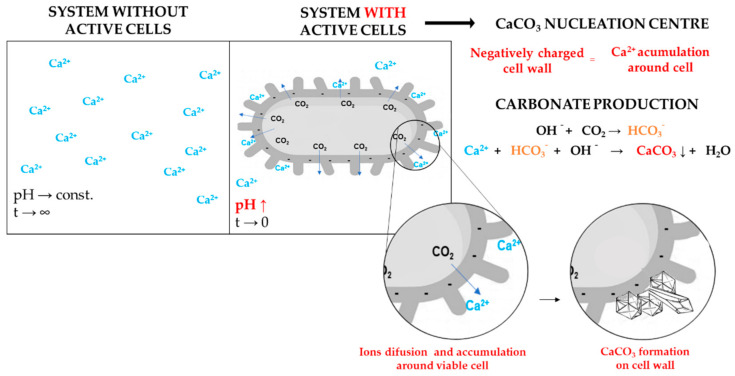
Bacterial cell as an active nucleation centre.

**Figure 2 microorganisms-10-01399-f002:**
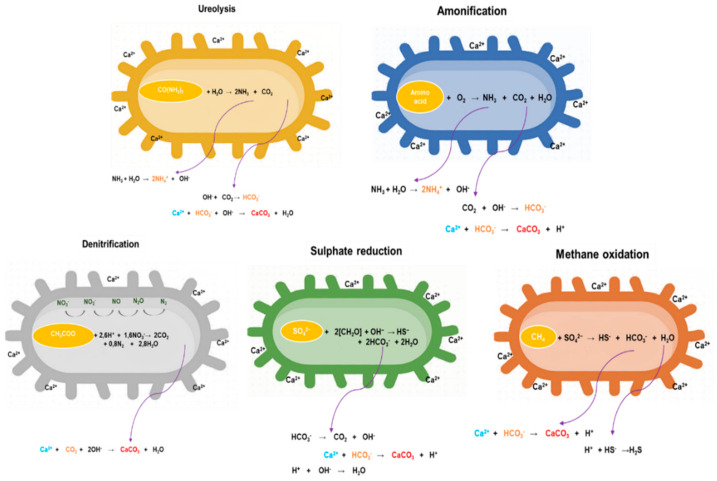
Metabolic pathways for effective bacterially induced carbonate precipitation.

**Figure 3 microorganisms-10-01399-f003:**
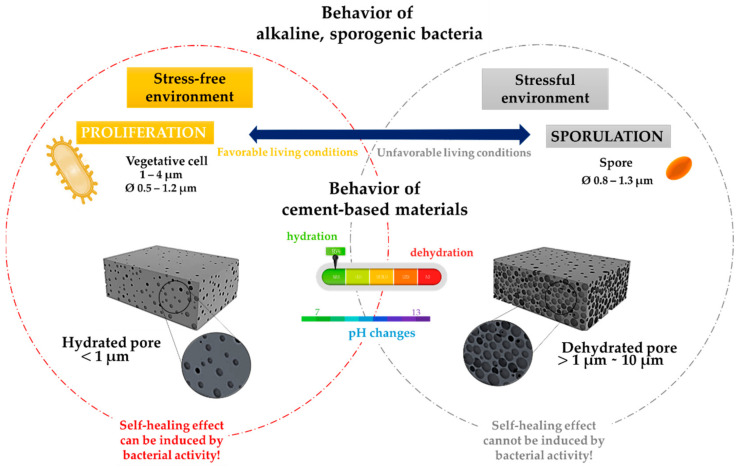
The bacterial life cycle depends on the cement-based material.

**Figure 4 microorganisms-10-01399-f004:**
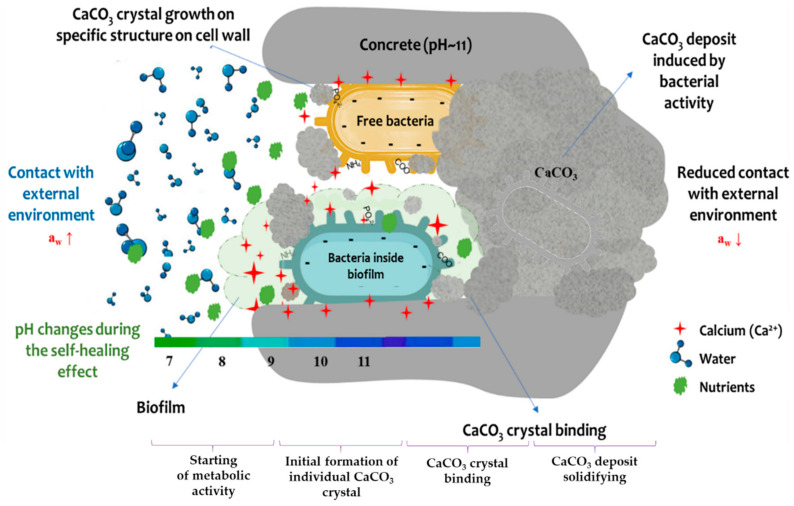
Bacterial behaviour in concrete.

**Figure 5 microorganisms-10-01399-f005:**
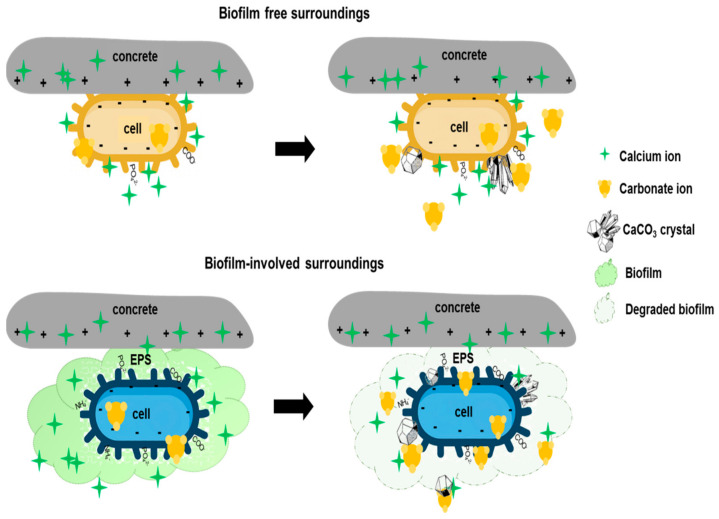
Differences in CaCO_3_ crystal production with and without biofilm.

**Figure 6 microorganisms-10-01399-f006:**
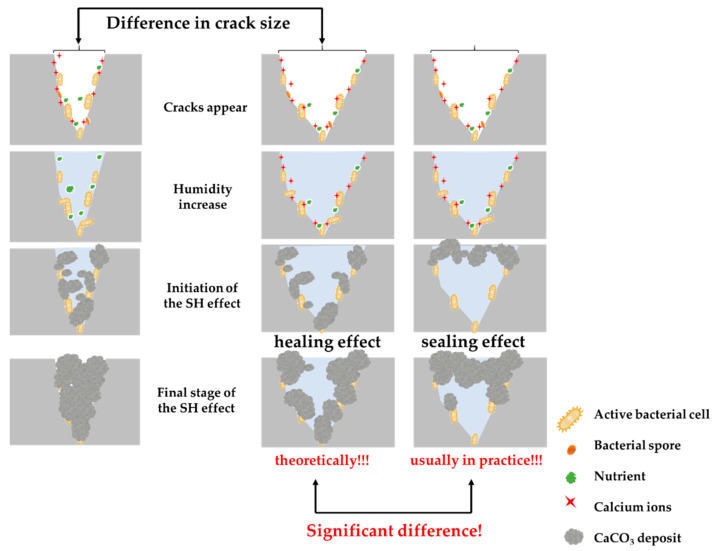
The difference in the SH effect depends on crack size.

**Table 1 microorganisms-10-01399-t001:** An overview of certain bacteria with BICP potential isolated from cement-based materials.

Isolation Site		
**Sediments**		
**Bacteria**		**Ref.**
*Sporosarcina pasteurii* *S. luteola* *Bacillus lentus*		[40]
*Brevundimonas dimitiuda*		[41]
*Arthrobacter* sp. *Flavobacterium* sp. *Pseudomonas* sp.		[42]
*B. pumilis* *P. grimonti* *Halomonas* sp.		[43]
*Lysinibacillus sphaericus*		[44]
*Enterobacter calcerogenus* *B. subtilis* *B. cereus*		[45]
*B. lentus* *B. fortis* *Sporosarcina* sp. *Pseudogracibacillus* sp.		[46]
*B. licheniformis* *B. muralis*		[47]
**Cement-Based Materials**		
**Bacteria**		**Ref.**
*B. sphaericus*		[48]
*Bacillus* sp. *Paenibacillus* sp. *Arthrobacter* sp.		[49]
*B. lentus* *B. sphaericus*		[50]
*B. thuringiensis* *B. pumilis*		[51]
*Bacillus* sp. *Brevibacillus* sp.		[52]
*P. azotoformanis*		[53]
*B. licheniformis*		[54]
*Bacillus* sp., *Sporosarcina* sp.		[55]
**Reference Strains**		
**Bacteria**	**Collection of Microorganisms**	**Ref.**
*Sporosarcina pasteurii* DSM 33 *B. cohnii* DSM 6307 *B. pseudofirmus* DSM 8715	DSM ^1^	[56,57]
*S. pasteurii* ATCC 11859	ATCC ^2^	[58]
*S. pasteurii* NCIM 2477	NCIM ^3^	[59]
*S. pasteurii* KCTC 3558	KTTC ^4^	[60]
*B. sphaericus* LMG 22257	LMG ^5^	[61]
*B. lentus* NCIB 8773	NCIB ^6^	[62]
*Myxococcus xanthus* CECT 422T	CECT ^7^	[41]
*B. mucilaginous* L3	CICC ^8^	[63]

^1^ DSM—German Collection of Microorganisms and Cell Cultures; ^2^ ATCC—American Type Culture Collection; ^3^ NCIM—National Collection of Industrial Microorganisms (India); ^4^ KCTC—Korean Collection for Type Cultures; ^5^ BCCM/LMG bacterial collection (Laboratory of Microbiology, Department of Biochemistry and Microbiology, Faculty of Sciences of Ghent University); ^6^ NCIB—National Collection of Industrial, Food and Marine Bacteria (UK); ^7^ CECT—Collection Nationale de Cultures de Microorganismes (France); ^8^ CICC—China Centre of Industrial Culture Collection.

**Table 2 microorganisms-10-01399-t002:** Selected examples of different approaches in view of bacteria-relevant parameters and monitoring of the self-healing (SH) system.

References			[81]	[56]	[82]	[83]	[84]	[85]	[20]	[86]	[87]
Type of Materials			Ordinary Portland Cement (OPC)								
**Bacteria**			Spore-forming alkali-resistant bacterium	*B. cohnii* DSM 6307 *B. halodurans* DSM 497 *B. pseudofirmus* DSM 8715	*B. mucilaginous L3*	*B. subtilis* jc3	*S. pasteurii* DSM 33	*B. subtilis*	*P. aeruginosa * *Diaphorobacter* *nitroreducens*	Anaerobic consortium (*Pseudomonas*, *Azotobacter*)	*B. pseudofirmus* *D. nitroreducens*
**Metabolic Activity**			Ammonification			Ureolysis			Denitrification		
**Vegetative cells (V) or spores (S)**			V	S	S	nd	V	V	V	V	V
**Initial bacterial concentration (CFU) ^a^**			10^9^	10^7^	10^10^	10^5^	10^7^	10^3^–10^9^	10^7^	10^8^	10^7^
**Monitoring of bacterial activity**			Nd ^b^	Assessment of spore activation	nd	nd	nd	nd	Checking viability ^c^	nd	nd
**Monitoring**	**self-healing effect**	Scanning electronic microscopy	+	−	+	−	+	+	−	+	+
		X-ray diffraction analysis	+	−	+	−	+	−	−	+	−
		Optical microscopy	−	−	+	+	−	−	+	+	+
		Chloride permeability	−	−	+	−	−	−	−	−	−
		Fourier−transform IR spectroscopy	−	−	−	−	+	−	−	−	−
		Surface resistivity	−	−	−	−	−	−	−	−	+
		Healing ratio ^d^	+	−	−	−	−	−	−	−	−
		CaCO_3_ precipitation potential	−	+	−	−	−	−	−	−	−
	**concrete performance**	Compressive strength	−	+	+	−	+	+	−	−	+
		Tensile strength	−	+	−	−	+	−	−	−	−
		Water permeability	+	−	+	+	+	−	−	+	−
		Water absorption	−	−	−	−	−	+	+	−	−
		Durability assessment	−	−	−	+	−	−	−	−	−
		Concrete density	−	−	+	−	−	−	−	−	−
		Ultrasonic pulse velocity	−	−	−	−	−	+	−	−	−
		Concrete slump test	−	−	+	−	−	−	−	−	−
		Setting time test	−	−	+	−	−	−	−	−	−
		Static modulus of elasticity	−	−	−	−	−	−	−	−	+

^a^ CFU—colony-forming unit; ^b^ nd—not determined; ^c^ after dehydration and starvation stress; ^d^ crack healing ratio using digital images setting—crack area threshold specific grey level.

**Table 3 microorganisms-10-01399-t003:** Changes in porosity and permeability of freshly prepared concrete.

Bacteria	Inoculation Level (CFU)	After the SH Effect ^a^		Healed Crack Width (μm)	Ref.
		Permeability	Porosity		
*B. mucilaginous* L3	10^10^	↓ *	nd **	300–500	[82]
*B. sphaericus* LMG 22257		↓	↓	200–900	[126]
Spore-forming alkali-resistant bacterium	10^9^	↓	↓	100–800	[81]
*B. cohnii* DSM 6307		↓	nd	1240	[127]
*B. sphaericus* LMG 22257		↓	↓	970	[5]
Anaerobic consortium	10^8^	↓	nd	100–1200	[86]
*Bacillus* sp. CT5		↓	↓	3000	[128]
*B. subtilis* 5265T		↓	↑ ***	400	[129]
*S. pasteurii* DSM 33	10^7^	↓	nd	200	[84]
*B. cereus* CS1		↓	nd	800	[130]
*B. subtilis* jc3	10^5^	↓	↓	200	[83]
*B. sphaericus* LMG 22257	nd	↓	nd	250–400	[131]

^a^—compared to control samples; * decrease; ** nd—not defined; *** increase.

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
