# Peer review of "Relationship between Bacterial Contribution and Self-Healing Effect of Cement-Based Materials"

_microorganisms, 2022, doi:10.3390/microorganisms10071399_

Round 1

Reviewer 1 Report

In this paper, the relationship between bacterial contribution and self-healing effect of cement-based materials is studied. The main problems in this paper are as follows:

1. What are the innovative points of this paper, which should be clearly presented in the abstract and introduction.

2. The clarity of the pictures in this article needs to be improved.

3. The references cited in this paper are old, it is suggested to quote more references in the past five years.

4. The format of this article still has some problems and needs to be modified.

5. Overall the author of this paper has done a good job and recommends that it is accepted after minor revision.

Author Response

Reviewer #1

In this paper, the relationship between bacterial contribution and self-healing effect of cement-based materials is studied.

ANSWER: The Authors would like to thank the Reviewer for a quick and professional review as well as the opportunity to make essential and crucial changes in our work. All the Reviewer' remarks are accepted and the paper is changed according to their comments. The Authors believe that the changed paper would satisfy the Reviewer' criteria and that it is going to be interesting enough for publishing in the Microorganisms.

We decided to revise the manuscript according to the Reviewer' remarks, highlighting the changes directly in the revised manuscript.

The main problems in this paper are as follows:

  1. What are the innovative points of this paper, which should be clearly presented in the abstract and introduction.

ANSWER: Thank you for this remark. The innovative points are presented in a better manner in the abstract and introduction.

  1. The clarity of the pictures in this article needs to be improved.

ANSWER: Thank you for this suggestion. All Figures are improved and better quality them are involved in the paper.

  1. The references cited in this paper are old, it is suggested to quote more references in the past five years.

ANSWER: Thank you for this suggestion. The references are refreshed, and app. 10 newest references are involved in the revised version (2019-2022). 

  1. The format of this article still has some problems and needs to be modified.

ANSWER: The manuscript was completely checked and formatted according to this suggestion.

  1. Overall the author of this paper has done a good job and recommends that it is accepted after minor revision.

ANSWER: Thank you for this comment, the Authors believe that the Reviewer had a strong impact on upgrading the whole paper for publication in the Microorganisms.

Reviewer 2 Report

There have been numerous reviews on self-healing concrete over the past 2 or 3 years. I think that this paper does provide new insights as it sits between the more pure microbiology and mechanistic reviews on microbial induced calcite precipitation, such as [1], and the many reviews that are more concrete application based. Therefore, I think the paper will be of interest to many readers.

Perhaps an aspect of the work that is missing, or would be worth adding, is reference to the environment in which the concrete healing is required. For example, there are some interesting aspects related to chloride environments [2], carbonation [3] and also the presence of sulfates [4].

Line 78 Figure 1. A supramolecular explanation for the negative cell wall and its role in calcite precipitation is given by Hoffmann [1] and the authors would do well to make it clear that they are aware of this review.

Line 183. Table 1. The authors should state in the caption that this is far from an exhaustive list.

Line 229. Authors state that “…bacterial survival was in the range of 1.9% and 7% after 10 days of curing”. It might be worth the authors stating their opinion on why this is. For example, I have seen some suggestions that it is down to pH but in my opinion it is more likely to be due to them being crushed by the hydrating cement. What is the authors’ view?

Line 378. It is true that not all calcium is available for self-healing. I think in this section the authors ought to have considered that it might in some circumstances be preferable to encapsulate the calcium source too. For example, see work at Delft where calcium lactate is encapsulated in lightweight aggregates [5].

Line 396. In nearly all cases the calcium salt, if added directly, will dissolve. One of the reasons for the success of calcium chloride is its high solubility. The result of this, however, is that abundant calcium hydroxide forms, and it seems that it is this calcium hydroxide that is the source of calcium used by the bacteria [6]. However, recent research has suggested that this can carbonate and be unavailable for future reactions, see [4].

Line 407. The discussion of ions in concrete in this section seems a little confused. 80% of the composition of concrete is normally aggregate and this has a largely mineral composition that is unreactive. The reactive part of the concrete is the cement paste. The composition of this depends on whether it is hydrated or not. The description given seems to refer to a non-hydrated paste, whereas a hydrated paste in concrete is primarily composed of calcium silicate hydrates, calcium hydroxide, calcium sulfoaluminate hydrates and other AFm phases. The other components mentioned are largely trace elements.  

Line 478. It is stated that “More porous concrete has a high rate of permeability”. The authors need to be careful here. Porosity and permeability are distinct characteristics of concrete; that are not necessarily related. For example, air entrainment can be used to provide concrete with freeze/thaw resistance without necessarily increasing the permeability of concrete. Provided pores are not well connected, permeability will not increase.

Line 486. I would have thought that the water for SH comes through the crack from the external environment. I’m not sure that pore water is sufficient.

Line 501. I don’t fully understand what the authors mean here. Permeability is a characteristic of hardened concrete whilst hydration of cement occurs before hardening. Consequently, the hydration of the cement defines the permeability and strength of the concrete. The effect of the permeability of hardened concrete on longer term hydration is likely to be minimal.

Line 515 Figure 6. I think these figures could be a little misleading. Nearly all current research suggests that healing takes place first at the surface. This healing at the surface then prevents ingress of oxygen leading to an oxygen deficit deeper in the crack and healing stops [7]. Consequently you end up with crack mouth sealing. Some partial healing at depth was suggested by Tan [8], perhaps. Indeed, work at Sydney showed, interestingly, that if the healing at the surface is not very good then, and only then, do you get good healing at depth [2]. So perhaps the secret is to try and make the healing slightly less efficient. Or alternatively microbial healing needs to be coupled with an inorganic method as described in recent work [9] to get healing at depth.

Line 531. It is not clear what is meant by aging. Concrete of course hardens with time, but this is not a negative thing. The authors in this section refer to early-age concrete cracks, but these seem to be almost the opposite to aging as they occur before the concrete has fully hardened.

Line 536. Authors state that “…obtained results indicate that the SH effect is better in the first 28 days”. However, the references used are for autogenous healing – where of course long-term healing is likely to be limited. For bacterial healing the situation is perhaps different. A very recent paper suggests that bacterial healing is still fine at 22 months [10].

Line 539. As above, 22 months rather than just 60 days has been demonstrated.

References used in review (not necessarily to be included in the paper)

[1]         T.D. Hoffmann, B.J. Reeksting, S. Gebhard, Bacteria-induced mineral precipitation: a mechanistic review, Microbiology. (2021). doi:10.1099/mic.0.001049.

[2]         M.B.E. Khan, L. Shen, D. Dias-da-Costa, Self-healing behaviour of bio-concrete in submerged and tidal marine environments, Constr. Build. Mater. 277 (2021) 122332. doi:10.1016/j.conbuildmat.2021.122332.

[3]         L. Tan, B. Reeksting, V. Ferrandiz-Mas, A. Heath, S. Gebhard, K. Paine, Effect of carbonation on bacteria-based self-healing of cementitious composites, Constr. Build. Mater. 257 (2020) 119501. doi:10.1016/j.conbuildmat.2020.119501.

[4]         I.M. Riad, A.A. Elshami, M.M.Y. Elshikh, Influence of concentration and proportion prepared bacteria on properties of self-healing concrete in sulfate environment, Innov. Infrastruct. Solut. 7 (2022) 1–16. doi:10.1007/s41062-021-00670-2.

[5]         E. Tziviloglou, V. Wiktor, H.M. Jonkers, E. Schlangen, Bacteria-based self-healing concrete to increase liquid tightness of cracks, Constr. Build. Mater. 122 (2016) 118–125. doi:10.1016/j.conbuildmat.2016.06.080.

[6]         M. Ksara, R. Newkirk, S.K. Langroodi, F. Althoey, C.M. Sales, C.L. Schauer, Y. Farnam, Microbial damage mitigation strategy in cementitious materials exposed to calcium chloride, Constr. Build. Mater. 195 (2019) 1–9. doi:10.1016/j.conbuildmat.2018.10.033.

[7]         S.D. Nielsen, K. Koren, K. Löbmann, M. Hinge, A. Scoma, K.U. Kjeldsen, H. Røy, Constraints on CaCO 3 precipitation in superabsorbent polymer by aerobic bacteria, Appl. Microbiol. Biotechnol. (2019). doi:https://doi.org/10.1007/s00253-019-10215-4 APPLIED.

[8]         L. Tan, X. Ke, Q. Li, S. Gebhard, V. Ferrandiz-Mas, K. Paine, W. Chen, The effects of biomineralization on the localised phase and microstructure evolutions of bacteria-based self-healing cementitious composites, Cem. Concr. Compos. 128 (2022) 104421. doi:10.1016/j.cemconcomp.2022.104421.

[9]         A. Wang, Q. Zhan, C. Fu, Y. Wang, J. Zhou, Study on improving the self-repairing effect of cement-based materials by microbial mineralization coupled with inorganic minerals, Case Stud. Constr. Mater. 17 (2022) e01279. doi:10.1016/j.cscm.2022.e01279.

[10]      I. Justo-Reinoso, B.J. Reeksting, A. Heath, S. Gebhard, K. Paine, Evaluation of Cyclic Healing Potential of Bacteria-Based Self-Healing Cementitious Composites, Sustainability. (2022) 1–15.

Author Response

Reviewer #2:

There have been numerous reviews on self-healing concrete over the past 2 or 3 years. I think that this paper does provide new insights as it sits between the more pure microbiology and mechanistic reviews on microbial induced calcite precipitation, such as [1], and the many reviews that are more concrete application based. Therefore, I think the paper will be of interest to many readers.

ANSWER: The Authors would like to thank the Reviewer for a quick and professional review as well as the opportunity to make essential and crucial changes in our work. All the Reviewer' remarks are accepted and the paper is changed according to their comments. The authors believe that the changed paper would satisfy the Reviewer' criteria and that it is going to be interesting enough for publishing in the Microorganisms.

We decided to revise the manuscript according to the Reviewer' remarks, highlighting the changes directly in the revised manuscript.

Perhaps an aspect of the work that is missing, or would be worth adding, is reference to the environment in which the concrete healing is required. For example, there are some interesting aspects related to chloride environments [2], carbonation [3] and also the presence of sulfates [4].

ANSWER: Thank you for very essential improvement of our review. We agree with this Reviewer’s suggestion, so we decided to upgrade Section 5 and involve some mentioned aspects of concrete environment conditions which can affect bacterial activity.

Line 78 Figure 1. A supramolecular explanation for the negative cell wall and its role in calcite precipitation is given by Hoffmann [1] and the authors would do well to make it clear that they are aware of this review.

ANSWER: Thank you for this suggestion, we added this reference and explained the connection between facts given in this work and our review study.

Line 183. Table 1. The authors should state in the caption that this is far from an exhaustive list.

ANSWER: Thank you for this comment, we changed the Table name.

Line 229. Authors state that “…bacterial survival was in the range of 1.9% and 7% after 10 days of curing”. It might be worth the authors stating their opinion on why this is. For example, I have seen some suggestions that it is down to pH but in my opinion it is more likely to be due to them being crushed by the hydrating cement. What is the authors’ view?

ANSWER: Thank you for this observation. We added our hypothesis to the text. We tried to identify the most important influence on cell activity in our laboratory. It is not a pH value for sure. If we select an appropriate sporogenic bacterium whose biokinetic zone of growth is in the range of concrete pH value, and the nutrients are involved in the system, dry/wet cycles have the most important influence on the number of viable cells. For sure, many operating and environmental conditions can have an additional impact, but it is different at the bacterial strain level. For example, urease-producing Bacillus strains are completely different in view of survival compared with non-ureolytic Bacillus strains. The same difference is observed in the comparison between Sporosarcina and B. licheniformis. We hope that we publish this article by the end of this year.

Line 378. It is true that not all calcium is available for self-healing. I think in this section the authors ought to have considered that it might in some circumstances be preferable to encapsulate the calcium source too. For example, see work at Delft where calcium lactate is encapsulated in lightweight aggregates [5].

ANSWER: Thank you for this suggestion. In the first four, five years in this field, we did not use the protection for nutrients, but after we tried to commercialized healing agent, we are dealing with the stability not only bacterial cells, but also nutrients for it. Therefore, we really appreciate this suggestion. We had this reference in review list, so we try to involve it in this part, but with some critical opinion at the end.

Line 396. In nearly all cases the calcium salt, if added directly, will dissolve. One of the reasons for the success of calcium chloride is its high solubility. The result of this, however, is that abundant calcium hydroxide forms, and it seems that it is this calcium hydroxide that is the source of calcium used by the bacteria [6]. However, recent research has suggested that this can carbonate and be unavailable for future reactions, see [4].

ANSWER: Thank you for this suggestion. We carefully read reference [4] and involve this hypothesis in the text

Line 407. The discussion of ions in concrete in this section seems a little confused. 80% of the composition of concrete is normally aggregate and this has a largely mineral composition that is unreactive. The reactive part of the concrete is the cement paste. The composition of this depends on whether it is hydrated or not. The description given seems to refer to a non-hydrated paste, whereas a hydrated paste in concrete is primarily composed of calcium silicate hydrates, calcium hydroxide, calcium sulfoaluminate hydrates and other AFm phases. The other components mentioned are largely trace elements.  

ANSWER: Thank you for this suggestion. We reorganized this section to better understand this influence in view of bacterial activity.

Line 478. It is stated that “More porous concrete has a high rate of permeability”. The authors need to be careful here. Porosity and permeability are distinct characteristics of concrete; that are not necessarily related. For example, air entrainment can be used to provide concrete with freeze/thaw resistance without necessarily increasing the permeability of concrete. Provided pores are not well connected, permeability will not increase.

ANSWER: Thank you for this suggestion, we rewrote this sentence.

Line 486. I would have thought that the water for SH comes through the crack from the external environment. I’m not sure that pore water is sufficient.

ANSWER: Thank you for this observation, we agree with this, the most bacteria require more than 0.95 aw for activity, so we changed this sentence.

Line 501. I don’t fully understand what the authors mean here. Permeability is a characteristic of hardened concrete whilst hydration of cement occurs before hardening. Consequently, the hydration of the cement defines the permeability and strength of the concrete. The effect of the permeability of hardened concrete on longer term hydration is likely to be minimal.

ANSWER: Thank you for this observation. We think about it, and decided to delete this part because of the double meaning in view of microbiology and civil engineering.

Line 515 Figure 6. I think these figures could be a little misleading. Nearly all current research suggests that healing takes place first at the surface. This healing at the surface then prevents ingress of oxygen leading to an oxygen deficit deeper in the crack and healing stops [7]. Consequently you end up with crack mouth sealing. Some partial healing at depth was suggested by Tan [8], perhaps. Indeed, work at Sydney showed, interestingly, that if the healing at the surface is not very good then, and only then, do you get good healing at depth [2]. So perhaps the secret is to try and make the healing slightly less efficient. Or alternatively microbial healing needs to be coupled with an inorganic method as described in recent work [9] to get healing at depth.

ANSWER: Thank you for this suggestion. We changed the Figure and involve some differences between theoretically and experimentally obtained results. We believe that this approach is the best chance to understand sealing moment in this field.

Line 531. It is not clear what is meant by aging. Concrete of course hardens with time, but this is not a negative thing. The authors in this section refer to early-age concrete cracks, but these seem to be almost the opposite to aging as they occur before the concrete has fully hardened.

Line 536. Authors state that “…obtained results indicate that the SH effect is better in the first 28 days”. However, the references used are for autogenous healing – where of course long-term healing is likely to be limited. For bacterial healing the situation is perhaps different. A very recent paper suggests that bacterial healing is still fine at 22 months [10]. Line 539. As above, 22 months rather than just 60 days has been demonstrated.

ANSWER: Thank you for this suggestion, we changed this part, and involve new information in order to demarcate autogenous and BICP effect and spectre of different investigation of early age and mature concrete samples.

References used in review (not necessarily to be included in the paper)

[1]         T.D. Hoffmann, B.J. Reeksting, S. Gebhard, Bacteria-induced mineral precipitation: a mechanistic review, Microbiology. (2021). doi:10.1099/mic.0.001049.

[2]         M.B.E. Khan, L. Shen, D. Dias-da-Costa, Self-healing behaviour of bio-concrete in submerged and tidal marine environments, Constr. Build. Mater. 277 (2021) 122332. doi:10.1016/j.conbuildmat.2021.122332.

[3]         L. Tan, B. Reeksting, V. Ferrandiz-Mas, A. Heath, S. Gebhard, K. Paine, Effect of carbonation on bacteria-based self-healing of cementitious composites, Constr. Build. Mater. 257 (2020) 119501. doi:10.1016/j.conbuildmat.2020.119501.

[4]         I.M. Riad, A.A. Elshami, M.M.Y. Elshikh, Influence of concentration and proportion prepared bacteria on properties of self-healing concrete in sulfate environment, Innov. Infrastruct. Solut. 7 (2022) 1–16. doi:10.1007/s41062-021-00670-2.

[5]         E. Tziviloglou, V. Wiktor, H.M. Jonkers, E. Schlangen, Bacteria-based self-healing concrete to increase liquid tightness of cracks, Constr. Build. Mater. 122 (2016) 118–125. doi:10.1016/j.conbuildmat.2016.06.080.

[6]         M. Ksara, R. Newkirk, S.K. Langroodi, F. Althoey, C.M. Sales, C.L. Schauer, Y. Farnam, Microbial damage mitigation strategy in cementitious materials exposed to calcium chloride, Constr. Build. Mater. 195 (2019) 1–9. doi:10.1016/j.conbuildmat.2018.10.033.

[7]         S.D. Nielsen, K. Koren, K. Löbmann, M. Hinge, A. Scoma, K.U. Kjeldsen, H. Røy, Constraints on CaCO 3 precipitation in superabsorbent polymer by aerobic bacteria, Appl. Microbiol. Biotechnol. (2019). doi:https://doi.org/10.1007/s00253-019-10215-4 APPLIED.

[8]         L. Tan, X. Ke, Q. Li, S. Gebhard, V. Ferrandiz-Mas, K. Paine, W. Chen, The effects of biomineralization on the localised phase and microstructure evolutions of bacteria-based self-healing cementitious composites, Cem. Concr. Compos. 128 (2022) 104421. doi:10.1016/j.cemconcomp.2022.104421.

[9]         A. Wang, Q. Zhan, C. Fu, Y. Wang, J. Zhou, Study on improving the self-repairing effect of cement-based materials by microbial mineralization coupled with inorganic minerals, Case Stud. Constr. Mater. 17 (2022) e01279. doi:10.1016/j.cscm.2022.e01279.

[10]      I. Justo-Reinoso, B.J. Reeksting, A. Heath, S. Gebhard, K. Paine, Evaluation of Cyclic Healing Potential of Bacteria-Based Self-Healing Cementitious Composites, Sustainability. (2022) 1–15.

ANSWER: Thank you for this comprehensive reference list, we tried to involve all suggestions and references in the text. Your contribution to our work is meaningful.
